# Effect of *Peganum harmala* Total Alkaloid Extract on Sexual Behavior and Sperm Parameters in Male Mice

**DOI:** 10.3390/vetsci10080498

**Published:** 2023-08-02

**Authors:** Hanane Derbak, Kálmán Imre, Amira Chahrazad Benabdelhak, Mohamed Moussaoui, Amina Kribeche, Rosa Kebbi, Abdelhanine Ayad

**Affiliations:** 1Department of Biological and Environmental Sciences, Faculty of Nature and Life Sciences, University of Bejaia, Bejaia 06000, Algeria; hanane.derbak@univ-bejaia.dz (H.D.); amirachahrazad.benabdelhak@univ-bejaia.dz (A.C.B.); mohamed.moussaoui@univ-bejaia.dz (M.M.); rosa_ecolo@hotmail.com (R.K.); 2Department of Animal Production and Veterinary Public Health, Faculty of Veterinary Medicine, University of Life Sciences “King Mihai I” from Timișoara, 300645 Timisoara, Romania; 3Pharmaceutical Sciences Research Center (CRSP), Constantine 25000, Algeria; 4Laboratory of Biomathematics, Biochemistry, Biophysics and Scientometrics (L3BS), University of Bejaia, Bejaia 06000, Algeria; amina.kribeche@univ-bejaia.dz

**Keywords:** *Peganum harmala*, sexual behavior, sperm parameters, testosterone, male mice

## Abstract

**Simple Summary:**

In recent years, special attention has been paid to studies on the effect of various plants on reproduction using laboratory animals. It is known that one of the major problems in a couple’s life is infertility and sperm dysfunction. Many species of plants in the Mediterranean area have been used in the food, pharmaceutical, and cosmetics industries, such as *P. harmala*, which represents one of several plants that constitute a source of natural substances. The results of this study indicate that the extract of *P. harmala* at a high concentration seems to have more of an effect on sexual behavior in male mice. In the current study, the high concentration of *P. harmala* alkaloid extract improved the sperm count in the treated mice in comparison with the control group. However, no difference was recorded in sperm motility in the treated groups. The results revealed that the membrane integrity of the spermatozoa seemed to be stable in the treated male. However, seminal fructose and testosterone concentrations slightly increased in all groups. Based on the results of this study, it seems that the extract of *P. harmala* growing in Algeria can affect some reproductive indices, such as the gonadosomatic index and sperm count.

**Abstract:**

The study was designed to evaluate the effects of the total alkaloid extract of Algerian *Peganum harmala* seeds on sexual behavior and male reproductive function. After two weeks of acclimatization, the male mice were randomly divided into four groups (seven mice in each group). For 35 days, the extract was administered orally at dose levels of 6.25, 12.5, and 25 mg/kg body weight per day to the respective groups of male mice (n = 7) and normal saline daily to the control group. On day 28, sexual behavior parameters were recorded. At the end of the trial, reproductive organ weights, sperm quality, seminal fructose, and testosterone hormone levels were evaluated. The three treated groups were compared with the control using statistical variance analysis (one-way ANOVA, *p* < 0.05), followed by Tukey’s test. The results of the groups treated with 12.5 and 6.25 mg/kg of *P. harmala* alkaloid revealed the MF and IF parameters to be the lowest compared to the control group (*p* < 0.05). However, the male mice treated with 25 mg/kg recorded the highest values. A low significant value of ML was observed in the group treated with 25 mg/kg of the total alkaloid extract of *P. harmala* compared to the control group (*p* < 0.01), while a rise was observed in the concentration group treated with 6.25 mg/kg. Regarding IL, the male mice treated with different concentrations of the total alkaloid extract of *P. harmala* recorded a higher time than the control group. Moreover, an increase in the gonadosomatic index was noticed in all groups compared to the control group. However, there was a significant (*p* < 0.01) decrease in the sperm counts of the groups treated with 12.5 mg/kg and 6.25 mg/kg. However, there was no significant difference in the motility, membrane integrity, and total antioxidant capacity of sperm cells compared to the control. The extract treatment also brought about a non-significant increase in fructose content of the seminal vesicle and serum testosterone level. The findings of this study demonstrate that the extract acts in a dose-dependent manner, and it has varying effects on the reproductive parameters of male mice.

## 1. Introduction

Since antiquity, people all over the world have used plants cultivated locally to maintain their health and improve their quality of life. Phytotherapy is based on the idea that plants possess natural properties that support the body and reduce ailments. For this reason, plants have been utilized as energizing supplements, vitalizing supplements, and potential enhancers of male sexual performance [1].

The World Health Organization defines infertility as a state where couples have not been able to conceive after a year of frequent sexual relations [1]. It has been reported that infertility is a common and complicated problem affecting 15% of all reproductive-aged couples [2], and about 50% of infertility cases are of male origin [3]. This decline is mainly related to incorrect lifestyles and environmental factors [4]. Furthermore, a scientific review of men’s semen quality using linear regression of data showed a significant decrease in average sperm count and seminal volume [5]. Therefore, the demand for natural aphrodisiacs is high, with an emphasis on improving the sexual performance of men, such as libido and penile erection.

The leaves, flowers, fruits, and seeds of several plants are known to possess bioactive compounds that act on physiological organs, particularly the reproductive system. A large number of medicinal plants have been used to treat sexual dysfunction and stimulate sexual activity in humans. For example, *Kaempferia parviflora* [6], *Ganoderma lucidum* [7], *Caralluma dalzielii* [8], *Saraca asoca* [9], *Carpolobia lutea* [10], and *Allium atroviolaceum* [11] have been shown to possess aphrodisiac properties, which improve a number of copulation parameters.

*Peganum harmala* L. (family Zygophyllaceae, called Harmel) is a vivacious plant, growing spontaneously in semi-arid environments, steppe regions, and sandy soils, especially in North Africa. Harmel seeds, bark, and roots have been used as folk medicine [12]. Several studies have shown that *P*. *harmala* contains natural substances, such as alkaloids, steroids, flavonoids, etc. [13]. Moreover, these seeds are the richest in compounds due to alkaloids’ structural diversity, especially β-carbolines such as harmine, harmaline, harmalol, and harmol, which have been proven to be the main substances responsible for their bioactivity [14]. Several investigations have reported a wide variety of pharmacological activity in *P. harmala* L. seeds, including anti-inflammatory [15], anti-fungal [16], insecticidal [17], antibacterial [18], and antidiabetic properties [19] *. P. harmala* also exhibits noticeable anticancer effects [20].

According to the literature, there have been few in vivo studies on the effect of seed extracts of *P. harmala* on male reproductive function, and the results are controversial [21,22,23,24]. None of these studies looked at sexual behavior. It should be noted that biochemical compounds can vary depending on the genetic factors, species varieties, cultivation or growth conditions, as well as geographical variations of the plant [25]. Additionally, the extraction method can alter the biochemical content [26]. To the best of our knowledge, the activity of *P. harmala* seed alkaloids growing in Algeria has been reported only on testicular histology [27]. The aim of the present study was to evaluate the in vivo activity of *P. harmala* seed extracts growing in Algeria on the sexual behavior and sperm quality of male mice (*Mus musculus*).

## 2. Materials and Methods

This study was performed in accordance with the guidelines for the care and purpose of laboratory animals. Ethics committee approval was received for the proposed experiments from the scientific committee of Faculty of Natural and Life Sciences, University of Bejaia (Report of Faculty Scientific Council #05 dated 14 December 2016).

### 2.1. Reagents and Chemicals

Estradiol benzoate and progesterone were procured from a local veterinary practice. All solvents and chemical products were obtained from Sigma Aldrich (Saint Louis, MO, USA).

### 2.2. Plant Materials

The seeds of *P. harmala* were collected in April 2016 from Ngaoues, Batna province, Algeria. The Global Positioning Systems location of this province is (35°32′ N, 6°10′ E). The seeds were identified by the botanist of the Bejaia University. The voucher specimen (PhB080) is deposited in the herbarium of the Pharmacy Department, University of Batna-2, Batna, Algeria. The seeds were dried under shade and then were ground to fine powder.

### 2.3. Preparation of Plant Extract

The alkaloid fraction was obtained by a recorded process. Briefly, *Peganum harmala* seeds, presented under a powdered (100 g) form, were extracted with ethanol (96%, *v*/*v*) in a Soxhlet apparatus for 12 h. The solvent from the ethanolic extract was completely removed and concentrated with a rotary evaporator. The obtained viscous extract was dissolved in HCl (2% *v*/*v*) and 500 mL petroleum ether, and the resulting apolar phase was removed. The pH value of the aqueous layer was basified with ammonia and extracted three times with chloroform (100 mL) [14]. The chloroform layer was evaporated under vacuum conditions in order to obtain approximately 0.05% *w*/*w* of alkaloid fraction.

### 2.4. Animal Preparation

In this experiment, sexually mature Swiss albino mice (8–10 weeks of age, 30–35 g, *Mus musculus*), of both sexes, were acquired from the animal house of the Central Faculty of the University of Constantine, Algeria. Animals were housed under controlled conditions (temperature 20 ± 4 °C with a 12 h light/12 h dark cycle). They were fed with pellets and tap water ad libitum. All experimental protocols were carried out according to the standard laboratory animal care regulations approved by the International Ethics Committee (Directive 2010/63/EU which updated and replaced the Directive of the European Council 86/609/EC). After two weeks of acclimatization, male mice were randomly divided into four groups (seven mice in each group). The control group received only carboxymethyl cellulose (CMC) vehicle (0.5%; 10 mL/kg). An amount of 6.25 mg/kg, 12.5 mg/kg and 25 mg/kg of *P. harmala* alkaloid fraction suspended in CMC (0.5%) per animal, per day, was administered in Group 1, Group 2 and Group 3, respectively. Experimental groups were treated over a period of 35 consecutive days. Twenty-eight female mice were sexually receptive and expressed behavior estrous after receiving subcutaneous injections of estrogen benzoate (10 mg/kg body weight) and progesterone (0.50 mg/kg body weight) at 48 h and 4 h, respectively.

### 2.5. Male Sexual Behavior Observation

After four weeks of the experience, male sexual behavioral tests were conducted during the dark phase (at 20:00 h). A single male mouse was placed in a transparent rectangular chamber (15 × 35 × 40 cm) under dim illumination, and allowed to acclimate for 10 min. Thereafter, a female stimulus-receptive was brought into the chamber. Sexual behavior parameters were monitored, including the following: mount frequency (MF) which establishes the number of mounts without intromission from the moment of female introduction to ejaculation; intromission frequency (IF) which establishes the number of intromissions from the moment of introduction of the female to the ejaculation; mount latency (ML) which establishes the time interval between introducing the female to the first mount by the male; intromission latency (IL) sets the period between introducing the female and the first intromission by the male.

### 2.6. Blood Sample and Gonadosomatic Index

At the end of the experiment, the male mice were sacrificed using chloroform anesthesia and the blood was collected by the cardiac puncture method. Then, the samples were immediately centrifuged (15 min at 1500× *g*), and the plasma was kept at −20 °C until hormonal dosing. Also, animals were subjected to laparotomy for removal and the weighing of the testes. The gonadosomatic index (GSI) was estimated using a formula that was determined by testis weight divided by body weight. The epididymis and seminal vesicle weight of each male mouse was taken.

### 2.7. Epididymal Sperm Count and Motility Analysis

Thereafter, the samples were incubated at 37 °C for 10 min to allow the sperm to disperse. A Malassez chamber was used to count the sperm, after a dilution of 1:10 with fixative (1 percent formalin in phosphate-buffered saline). The number of spermatozoa in the squares of the hemocytometer was counted under a light microscope at 40× magnification. In order to assess the motility of spermatozoa, 10 μL of sperm from each sample was deposited in a prewarmed Makler chamber (Sefi-Medical Instruments Ltd., Biosigma S.R.L., Cona, Italy). At least 200 sperm were evaluated per digital camera mounted on a Nikon Eclipse E200 microscope (Nikon, Tokyo, Japan), and then classed as mobile or immobile. The ratio of the number of motile sperm out from the total number of sperm was used to indicate their motility.

### 2.8. Membrane Integrity and Antioxidant Status of Spermatozoa

The hypo-osmotic swelling (HOS) test was performed to assess the sperm membrane integrity, as was previously described by [28]. Briefly, 100 µL of sperm was incubated for an hour at 37 °C with 1 mL of HOS solution (4.9 g sodium citrate and 9 g fructose in 1000 mL of distilled water). Two hundred spermatozoa were counted, and sperms with coiled tails were considered to have functional membranes. 

The total antioxidant capacity (TAC) of the samples was measured after the sperm sonication. It was estimated using ABTS (2,20-Azino-di-[3-ethylbenzthiazo- line sulphonate]). The method is based on the sperm’s antioxidant capacity to inhibit oxidation of ABTS (2,20-Azino-di-[3-ethylbenzthiazo- line sulphonate]) on ABTS+ [29].

### 2.9. Testosterone and Fructose Assay

Serum testosterone was measured using a reader of enzyme immunoassays and according to the standard protocol provided in the test kit (Human Diagnostics Worldwide: ELISA Testosterone, direct REF: 55010, Wiesbaden, Germany). Each sample was analyzed in duplicate. The detecting antibody was the rabbit anti-testosterone IgG, as biotin conjugate. The enzyme substrate was the avidin-horseradish peroxidase. The fructose content in the seminal vesicles was estimated spectrophotometrically using the resorcinol method as previously mentioned by [30].

### 2.10. Statistical Analysis

The recorded data were statistically analyzed using Graph Pad Prism 5.0 (San Diego, CA, USA) software. The results were expressed as mean ± standard error (mean ± SEM). The different treated groups were compared with the control using statistical variance analysis (one-way ANOVA), followed by Tukey’s test. Values were considered significant when *p* < 0.05.

## 3. Results

The effects of the total alkaloid extract of *P. harmala* on the sexual behaviors of male mice is illustrated in Figure 1. A significant low value in the mount latency was observed in the group treated with 25 mg/kg of the total alkaloid extract of *P. harmala* compared with the control group (*p* < 0.01), and a non-significant decrease (*p* > 0.05) in the groups treated with 12.5 mg/kg. On the other hand, groups treated with 6.25 mg/kg extract displayed an increased mount-latency trend. The mount frequency was significantly lower in the groups exposed to 12.5 mg/kg (*p* < 0.05) and 6.25 mg/kg (*p* < 0.01) of extract. However, it showed no significant improvement (*p* > 0.05) in the group exposed to 25 mg/kg of total alkaloid extract of *P. harmala* compared to the control. Regarding the intromission latency, a significant raise (*p* < 0.01) was observed in time of male mice exposed to the dose of 6.25 mg/kg, whereas the other treated groups showed a non-significant increase (*p* > 0.05) compared with the control group. The intromission frequency of the group exposed to the extract with a dose of 25 mg/kg increased non-significantly (*p* > 0.05), whereas male mice exposed to the concentration of 12.5 mg/kg showed a non-significant decrease (*p* > 0.05). The group treated with 6.25 mg/kg of *P. harmala* extract showed a significant reduction in intromission frequency compared with the control group (*p* < 0.001).

The administration of the total alkaloid *P. harmala* extract resulted in an increase in epididymal weight in case of the treated groups, but only the administration of 12.5 mg/kg dose showed a significant increase. In contrast, seminal vesicle weights in male mice did not change significantly after treatment (Table 1). However, there were significant (*p* < 0.001) increases in the gonadosomatic index in all groups treated with *P. harmala* extract compared to the control group.

The mean of sperm counts and sperm motility according to the used concentrations of the total alkaloid extract of *P. harmala* in the treated male mice are shown in Figure 2 and Figure 3. The administration of the 25 mg/kg concentration of *P. harmala* alkaloid extract resulted in an increase in sperm count in case of the treated groups in comparison with the control. The mean of the sperm count decreased significantly in case of the groups treated with 12.5 and 6.25 mg/kg concentrations compared to the control group (*p* < 0.01). However, no significant increase in sperm motility was observed in all groups of *P. harmala*-extract-treated mice compared to the control group.

The results of the membrane integrity of the spermatozoa seem to be stable in male mice treated with different concentration of *P. harmala* extract alkaloid and were practically similar with those recorded in the case of control group (Figure 4). The male mice treated with 25 mg/kg concentration of *P. harmala* extract showed a slight increase, but which was non-significant, of the sperm’s total antioxidant capacity compared to the control group (Figure 5).

The mean of seminal fructose and testosterone levels in blood serum have increased in the case of all groups treated with the total alkaloid *P. harmala* extract compared to the control group (Table 2). However, this rise was not statically significant.

## 4. Discussion

It is well known that plants play a crucial role in regulating people’s health and well-being. There is a growing interest in the consumption of plants for the benefit of people’s health. However, their use must be accompanied by research aimed to investigate their safety, particularly on the reproductive system. Traditional medicine has recourse to several aphrodisiac plants such as *P. harmala*. Nevertheless, the results of studies on the effects of the *P. harmala* extract on male reproduction and physiology are controversial.

The present investigation was undertaken to evaluate the effect of the total alkaloid extract of *P. harmala* on the sexual behavior and sperm parameters of male mice. The administration of *P. harmala* extract has revealed a significant effect on sexual behavior in male mice in comparison with the control group. High concentrations of the alkaloid extract increased ML and IL, and decreased MF and IF in male mice, respectively. The aphrodisiac herbs are defined as plants which exhibit a significant increase in the mount and intromission frequency (as indicators of sexual capacity), and a significant decrease in the mount and intromission latency (as indicators of sexual motivation) [31,32]. Several studies demonstrated the significant decrease in intromission and mounting latency, and the remarkable increase in intromission and mounting frequency, as a consequence of the administration of extract plants such as *Eremomastax speciosa* [33], *Hygrophila spinosa* [34] and *Dracaena arborea* [35]. However, in a recent study conducted by Obiandu and Achinike [36] it has been shown that the MF, ML, IF and IL parameters were not significantly altered after a 30-day of treatment with a dose of 200 mg/kg and 400 mg/kg extract of *M. charantia*. It was concluded that this plant may not affect sexual activity in male Wister rats within the administered doses.

The increase or decrease in sexual arousal is generally controlled by the brain, especially by the peripheral nervous system. It is also well known that the neurotransmitters in the brain are involved in modulation of sexual behavior [34]. In the present experiment, the modification in the sexual performance of male mice could be due to the action of the bioactive compounds of *P. harmala* alkaloid extract on the neurotransmitters rich in harmaline, harmine, harmalol and harmol. It has been reported that serotonin is an inhibitory neurotransmitter, whereas dopamine is recognized as an excitable neurotransmitter [37]. Dopamine is responsible for the motivational aspect of sexual behavior and it is related to desire [38]. Moreover, dopaminergic action enhances genital responses and sexual activity. Also, the effect of dopamine in Parkinson’s patients has been shown to provide experimental proof for the involvement of brain dopamine in human sexual behavior [39]. It is known that the majority of β-carboline alkaloids could inhibit the metabolism of catecholamine neurotransmitters [40]. Indeed, the effects of β-carbolines on dopaminergic transmission have been the subject of neurochemical and behavioral studies. These have shown that certain β-carboline alkaloids can facilitate dopaminergic transmission and interact with dopamine D1 and D2 receptors in the brain [41]. In addition, numerous neurotransmitters interact with dopamine, particularly monoamine neurotransmitters that are important for its stimulation [42]. However, the β-carboline alkaloids possess an inhibitory action on monoamine oxidase [43,44].

On the other hand, our results showed an increase in the weight of the epididymis and the GSI in male mice treated with *P. harmala* extracts (at 12.5 mg/kg concentration) which is in concordance with previous findings published by several authors [21,22,23,24,25,26,27,28,29,30,31,32,33,34,35,36,37,38,39,40,41,42,43,44,45]. This could be explained by some anatomical modifications in the testes, including hypertrophy of the seminiferous tubules, organization of the seminiferous epithelial cells, and a decrease in the number or degeneration of spermatogenic and Leydig cells, and a significant development of the epididymis tissue structure with a heavy sperm load. However, other researchers reported a significant decrease in the weight of reproductive organs [22,23,24]. This divergence of results could be due to the experimental conditions, such as the solvent or the used plant part to prepare the extract, the nature and the concentration of the extract, and the duration of the study, or the administration mode of extract. The weight gain of the sexual organs is generally under the control of androgens, which are required for the development, growth and normal function of male testes and sex glands. It is well known that testosterone is a steroid anabolic promoting protein synthesis. After conversion of the testosterone into dihydrotestosterone (DHT) by the -5 reductase enzyme, it subsequently attaches to cytoplasmic protein receptors and reaches the nucleus to promote the transcription of DNA into RNA. Therefore, muscle growth and the increase in bodyweight and the reproductive organs are all stimulated by the translation of RNA into protein [46].

In this study, the results of the treated mice with high doses of *P. harmala* total alkaloid extract (25 mg/kg) revealed a significant increase (*p* < 0.05) in sperm counts, whereas the results of motility and membrane integrity of the sperm does not seem to be affected by the different concentrations of the *P. harmala* total alkaloid extract compared to the control group. Our results are in accordance with those previously published, which showed that *P. harmala* extract caused a non-significant decrease in the concentration and percentage of motile spermatozoa [24]. Similar observations were reported by El-Dwairi and Banihani [22] in rats treated for 60 days with *P. harmala* seeds extract. In another study, the research team members concluded that the evaporation of *P. harmala* seeds for 7, 14, and 21 days, respectively, caused an increased sperm count in male rats treated with chloropromazine [47]. Likewise, the in vitro addition of low concentrations of total alkaloids *P. harmala* seeds in the sperm produced stimulation of motility and protection against oxidative damages [48]. This effect can also be related to the mineral composition of the tested plant, such as copper, manganese, and zinc, which are recognized as an important class of antioxidants [49]. For example, zinc leads to improvement in sperm motility because of its involvement in protein synthesis and nuclear chromatin stabilization [50]. Also, the calcium (Ca^2+)^ pathway and the cyclic adenosine monophosphate (cAMP)-dependent protein kinase are considered two important metabolic pathways involved in the regulation of sperm motility [51].

The antispermatogenic activity of several plant extracts have been previously observed on rat testes, resulting in a decline in sperm count [52,53]. Also, the plant extract may induce structural disturbances at the caput and cauda level of the epididymis, resulting in a reduction in epididymal sperm count [54]. It may alter the motility and membrane integrity of sperm. This could be explained by the desensitization of the sperm membrane receptors. Considering the fundamental role of the epididymis in sperm maturation and capacitation at the epididymis-tract level, the structural alterations in the epididymis can considerably affect sperm motility [55]. The presence of oxidative stress has been observed to be an important cause of idiopathic male infertility, which is a result of an imbalance between the production of reactive oxygen species (ROS) and their neutralization or scavenging by the antioxidant system [56]. It has been reported that a low antioxidant status is associated with a poor and damaged sperm quality, which may even affect the DNA structure, thus contributing to male infertility [57]. Furthermore, the total antioxidant status of semen can be considered as a valuable marker of sperm quality. In recent years, natural antioxidants obtained from plant extracts have been investigated in order to decrease semen inhibition by scavenging initiating radicals, and decomposing peroxides. Thus, Sarbishegi et al. [58] reported that olive leaf extract, after 30 days of administration, significantly improved the TAC and decreased the elevation of malondialdehyde (MDA) in the treated groups (*p* < 0.05). Also, Hosseinzadeh Colagaret al. [59] demonstrated a correlation between the lower levels of TAC and low sperm count, as well as their motility in the seminal plasma of asthenoteratozoospermic and oligoasthenoteratozoospermic men. The present study revealed a slight improvement in TAC of the sperm of male mice treated with alkaloid extracts of *P. harmala*. Numerous studies observed a positive correlation between the seminal plasma TAC and spermatic parameters, such as sperm concentration, sperm motility, and normal sperm morphology [60]. This suggests that a decreased level of seminal TAC could play a key role in the etiology of impaired sperm functions.

Fructose is secreted by the seminal vesicles and the accessory sex glands. It is essential for sperm viability and motility, serving as an energy source for spermatozoa [61]. The biosynthesis of fructose is controlled by androgens, and is therefore directly linked to testosterone concentrations [62]. Also, there is a direct relationship between the fructose level in sperm plasma and the testosterone function of Leydig interstitial cells. It has been reported that fructose determination is a suitable marker for the secretory function of the accessory reproductive gland. In the present study, seminal fructose levels and testosterone concentrations of the treated groups with *P. harmala* extract were higher compared with the control. It has been previously demonstrated that the quantity of the synthesized testosterone is indirectly controlled by the hypothalamus–pituitary–testicular axis through the releasing of FSH and LH hormones [63]. Several authors have reported that the enhancement of sexual behavior using medicinal plants was associated with an increase in testosterone levels [64,65,66]. In this current investigation, the alkaloids contained in *P. harmala* extract could exercise a direct and/or indirect effect on the hypothalamus–pituitary–gonadal axis; this may also explain their aphrodisiac activity.

## 5. Conclusions

In conclusion, this study’s results showed that *P. harmala* seed extracts, growing in Algeria, at 25 mg/kg concentration, affect the sexual performances (MF, ML and IF) and sperm count of male mice. Our results revealed no significant effect of the *P. harmala* seed extracts on sperm membrane integrity and sperm total antioxidant capacity. Fructose levels and testosterone concentrations tended to be higher in mice treated with *P. harmala* seed extract than in the mice from the control group. From these findings, it is concluded that *P. harmala* seed extracts did not sufficiently improve the reproductive function in male mice, thus suggesting a limitation of its use in male subjects. Nevertheless, more investigations are desired to test the reproductive effect of *P. harmala* of other geographic origins and to document the biochemical mechanism of action of the *P. harmala* total alkaloid extract in other cellular models.

## Figures and Tables

**Figure 1 vetsci-10-00498-f001:**
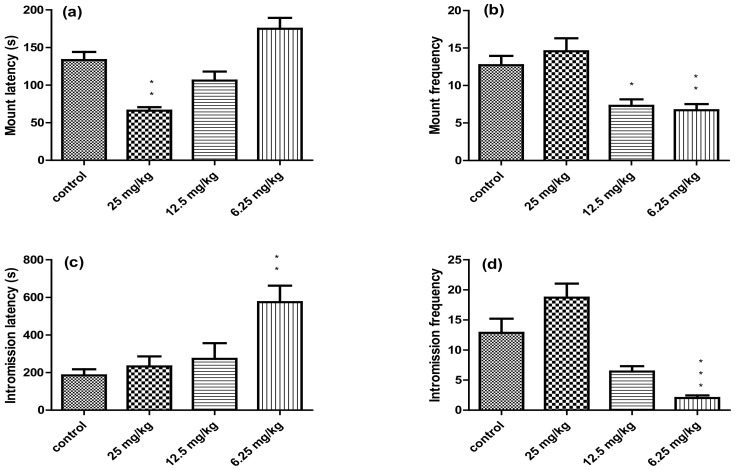
Results of sexual behavior of the control group and the treated groups with the total alkaloid extract of *Peganum harmala*: (**a**) latency of mount; (**b**) latency of intromission; (**c**) frequency of mount; (**d**) frequency of intromission. The data are expressed as means ± SEM. * *p* < 0.05; ** *p* < 0.01; and *** *p* < 0.001 compared with control group.

**Figure 2 vetsci-10-00498-f002:**
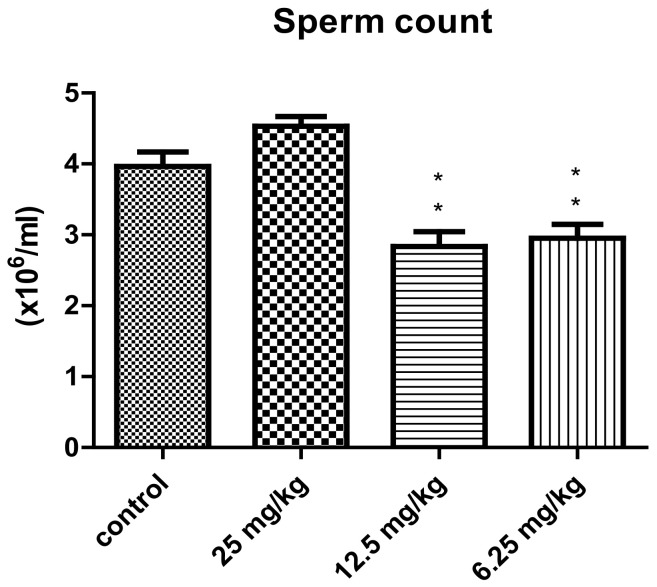
Effect of treatment with different concentrations of the total alkaloids extract of *Peganum harmala* on sperm count of male mice (mean ± SEM). ** *p* < 0.01 compared with control group.

**Figure 3 vetsci-10-00498-f003:**
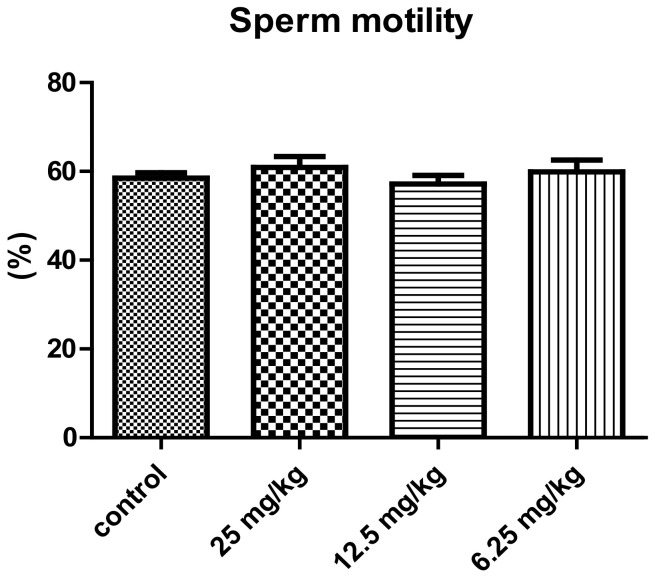
Effect of treatment with different concentrations of the total alkaloid extract of *Peganum harmala* on the percentage of sperm motility of male mice (mean ± SEM).

**Figure 4 vetsci-10-00498-f004:**
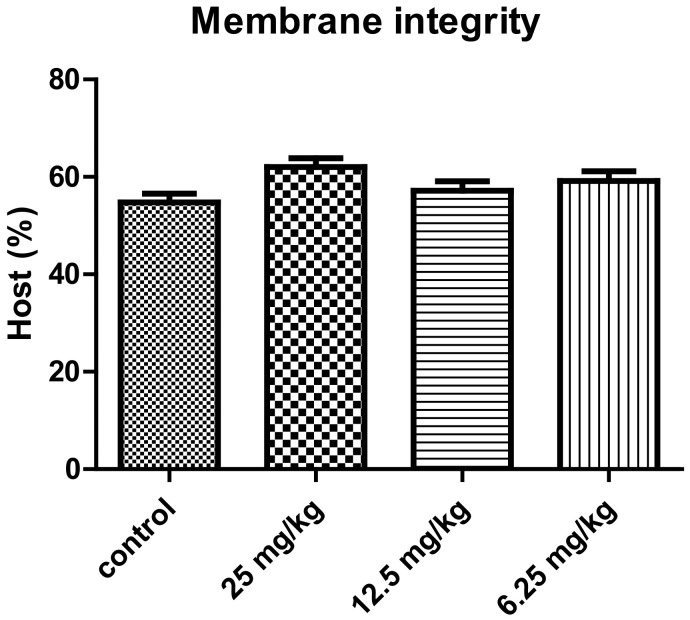
Effect of treatment with different concentrations of the total alkaloid extract of *Peganum harmala* on membrane integrity of sperm of male mice (mean ± SEM).

**Figure 5 vetsci-10-00498-f005:**
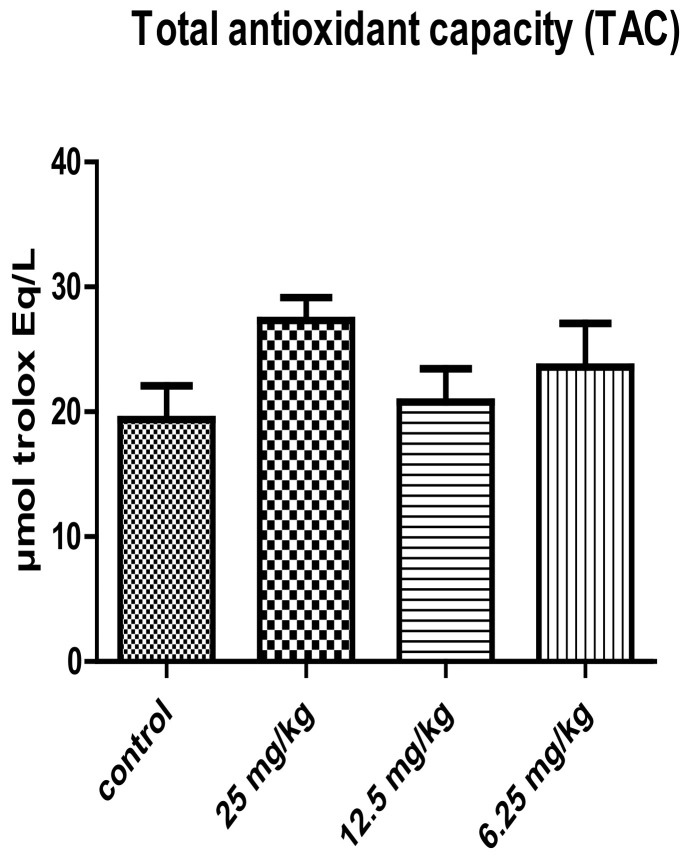
Effect of treatment with different concentrations of the total alkaloid extract of *Peganum harmala* on the TAC of sperm of male mice (mean ± SEM).

**Table 1 vetsci-10-00498-t001:** Effect of treatment with different concentrations of the total alkaloids extract of *Peganum harmala* on GSI and reproductive organ weights (g) of male mice (mean ± SEM,).

Experimental Groups				
	Control	25 mg/kg	12.5 mg/kg	6.25 mg/kg
GSI	0.140 ± 0.13	0.985 ± 0.06 ***	1.11 ± 0.06 ***	0.889 ± 0.04 ***
Epididymis weight (g)	0.063 ± 0.001	0.067 ± 0.005	0.082 ± 0.004 *	0.064 ± 0.004
Seminal vesicle weight (g)	0.181 ± 0.01	0.156± 0.01	0.183 ± 0.01	0.168 ± 0.01

* *p* < 0.05, *** *p* < 0.001 compared with control group.

**Table 2 vetsci-10-00498-t002:** Effect of treatment with different concentrations of the total alkaloids extract of *Peganum harmala* on seminal fructose concentration and serum testosterone level of male mice (mean ± SEM).

Experimental Groups				
	Control	25 mg/kg	12.5 mg/kg	6.25 mg/kg
Seminal fructose (mg/g)	3.72 ± 0.38	4.22 ± 0.5	4.61 ± 0.68	4.62 ± 0.79
Testosterone (ng/mL))	1.14 ± 0.39	2.08 ± 0.72	1.21 ± 0.55	1.37 ± 0.79

## Data Availability

Data are contained within the article.

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
