# Peer review of "Effect of Peganum harmala Total Alkaloid Extract on Sexual Behavior and Sperm Parameters in Male Mice"

_vetsci, 2023, doi:10.3390/vetsci10080498_

Round 1

Reviewer 1 Report

This study “Effect of Peganumharmala total alkaloid extract on sexual behavior and sperm parameters in male mice”  focuses on evaluating the effects of the total alkaloid extract of Peganum harmala Algerian seeds on sexual behavior and male reproductive function in male mice. The study assessed sexual behavior parameters such as mount latency, intromission latency, mount frequency, and intromission frequency. However, upon critically analyzing the manuscript, I have following observations regarding the study.

The study lacks the generalizability and clinical significance. The abstract of the study is not well justified as does not mention the sample size used in the study, making it difficult to evaluate the statistical power and reliability of the results. Additionally, the statistical analysis methods employed are not described, which may affect the robustness of the conclusions drawn. Further the results are not explicitly described i.e. the authors have not mentioned that what results were observed regarding mount latency (ML), intromission latency (IL), mount frequency (MF), intromission frequency (IF).

The introduction fails to clearly state the specific research gap or knowledge deficit that the study aims to address. While it mentions that there are few in-vivo studies on the effect of P. harmala seed extracts on male reproductive function, it does not explain why further investigation is necessary or how the current literature falls short. The authors in introduction section, mentions that the study focuses on P. harmala seeds growing in Algeria, but it does not provide a rationale for this geographical focus. Additionally, the introduction of the manuscript lacks a clear and compelling argument to convince the reader of the study's significance. It fails to provide a strong rationale for investigating sexual behavior, does not clearly define research objectives, and does not sufficiently address previous contradictory findings.

The methodology does not clearly state the specific research question or objective of the study. It briefly mentions evaluating the "pharmacological activity of P. harmala seed extracts growing in Algeria on sexual behavior and sperm quality in male mice," but there is no clear statement. The methodology states that only seven mice were included in each group. This small sample size may not provide sufficient statistical power to draw reliable conclusions or generalize the findings to a larger population. Larger sample sizes are generally preferred to improve the robustness and reliability of the results. Furthermore, does not mention whether blinding or randomization procedures were employed during the experiments. Blinding is important to minimize bias, while randomization helps ensure that the groups are comparable and reduces the risk of systematic errors. Although the manuscript suggests that the impaired sexual performance in male mice could be attributed to the action of bioactive compounds in the alkaloid extract, the specific mechanisms underlying these effects are not thoroughly explored or supported by experimental evidence. Without a clear understanding of the mechanisms involved, the significance of the findings may be limited. Moreover, the discussion of manuscript does not discuss the potential implications or applications of the findings in a clinical context, limiting the practical relevance of the study. The conclusion states that P. harmala seed extracts did not improve some parameters of male mice reproductive function, suggesting a limitation in its use in male subjects. However, the study does not provide a clear explanation or discussion regarding why these parameters were not improved and whether the observed effects are clinically significant or meaningful. Further interpretation and context are required to understand the practical implications of the findings.

Author Response

Response to Reviewer #1 Comments

This study “Effect of Peganum harmala total alkaloid extract on sexual behavior and sperm parameters in male mice”  focuses on evaluating the effects of the total alkaloid extract of Peganum harmala Algerian seeds on sexual behavior and male reproductive function in male mice. The study assessed sexual behavior parameters such as mount latency, intromission latency, mount frequency, and intromission frequency. However, upon critically analyzing the manuscript, I have following observations regarding the study.

Point 1: The study lacks the generalizability and clinical significance. The abstract of the study is not well justified as does not mention the sample size used in the study, making it difficult to evaluate the statistical power and reliability of the results. Additionally, the statistical analysis methods employed are not described, which may affect the robustness of the conclusions drawn. Further the results are not explicitly described i.e. the authors have not mentioned that what results were observed regarding mount latency (ML), intromission latency (IL), mount frequency (MF), intromission frequency (IF).

Response 1:

Certainly, it is crucial when conducting clinical studies, particularly randomised controlled trials, to produce data that may be used in clinical practice. Generalizability is a long-standing issue when applying trial results to patients in the real world, though. Although generalizability assessment is crucial, it is not always used. The designed clinical research studies, especially randomized controlled trials (RCTs), could allow regulatory agencies to approve new therapies and care providers to make better clinical decisions. However, an excessive focus on internal validity may result in the exclusion of specific population subgroups and poor generalizability.

The notions of generalizability and population representativeness are distinct but closely related. In clinical trials, three essential populations of interest exist: 1) the target population representing patients to whom the study results are intended to be applied in real-world patients; 2) the study population displaying the patients who are eligible for the study (based on study inclusion/exclusion criteria); and 3) the study sample representing participants who are enrolled in the clinical study. Generalizability is the ultimate portability of the causal effects of an intervention (developed based on the Study Sample) to the target population. In our study the main purpose was to evaluate the effectiveness of Peganum harmala total alkaloid extract on sexual behaviour for pharmacological purposes. This could be a gateway to further research in this area and subsequent generalisation. For this reason we have not opted for generalizability evaluation at this early stage. Other, more in-depth research in this area is currently being carried out to ensure more visibility of male mice behaviour towards the studied extract.

The methodology states that only seven mice were included in each group. This small sample size may not provide sufficient statistical power to draw reliable conclusions or generalize the findings to a larger population. Larger sample sizes are generally preferred to improve the robustness and reliability of the results. Furthermore, does not mention whether blinding or randomization procedures were employed during the experiments. Blinding is important to minimize bias, while randomization helps ensure that the groups are comparable and reduces the risk of systematic errors.

Point 2: The introduction fails to clearly state the specific research gap or knowledge deficit that the study aims to address. While it mentions that there are few in-vivo studies on the effect of P. harmala seed extracts on male reproductive function, it does not explain why further investigation is necessary or how the current literature falls short. The authors in introduction section, mentions that the study focuses on P. harmala seeds growing in Algeria, but it does not provide a rationale for this geographical focus. Additionally, the introduction of the manuscript lacks a clear and compelling argument to convince the reader of the study's significance. It fails to provide a strong rationale for investigating sexual behavior, does not clearly define research objectives, and does not sufficiently address previous contradictory findings.

Response 2:

We agree with you. Therefore, we have added the mean motivation of the present study in the section "Introduction". Please, see the revised version. We think it is clearer now.

It is intersecting to note the biochemical compounds variations may be due to genetic factor, varieties, culture or growth conditions, as well as geographical variations of plant (De Wit et al., 2018). Also, the stage of flower maturity can affect in molecular compounds of vegetal. In addition, the extraction method can alter biochemical content (Yeh et al., 2014). These are some arguments of the present investigation. It is assumed that the P. harmala growing in Algeria would be different compared to other works.

De Wit M., Hugo A., Shongwe N. South African cactus pear seed oil: A comprehensive study on 42 spineless burbank Opuntia ficus-indica and Opuntia robusta cultivars. European Journal of Lipid Science and Technology. 2018. T. 120(3). P. 1700343.

Yeh, H. -y., Chuang, C. -h., Chen, H. -c., Wan, C. -j., Chen, T. -l., & Lin, L. -y. (2014). Bioactive components analysis of two various gingers (Zingiber officinale Roscoe) and antioxidant effect of ginger extracts. LWT — Food Science and Technology, 55, 329–334

Point 3: The methodology does not clearly state the specific research question or objective of the study. It briefly mentions evaluating the "pharmacological activity of P. harmala seed extracts growing in Algeria on sexual behavior and sperm quality in male mice," but there is no clear statement. The methodology states that only seven mice were included in each group. This small sample size may not provide sufficient statistical power to draw reliable conclusions or generalize the findings to a larger population. Larger sample sizes are generally preferred to improve the robustness and reliability of the results. Furthermore, does not mention whether blinding or randomization procedures were employed during the experiments. Blinding is important to minimize bias, while randomization helps ensure that the groups are comparable and reduces the risk of systematic errors. Although the manuscript suggests that the impaired sexual performance in male mice could be attributed to the action of bioactive compounds in the alkaloid extract, the specific mechanisms underlying these effects are not thoroughly explored or supported by experimental evidence. Without a clear understanding of the mechanisms involved, the significance of the findings may be limited. Moreover, the discussion of manuscript does not discuss the potential implications or applications of the findings in a clinical context, limiting the practical relevance of the study. The conclusion states that P. harmala seed extracts did not improve some parameters of male mice reproductive function, suggesting a limitation in its use in male subjects. However, the study does not provide a clear explanation or discussion regarding why these parameters were not improved and whether the observed effects are clinically significant or meaningful. Further interpretation and context are required to understand the practical implications of the findings.

Response 3:

We would like to thank you for this comment, it is very pertinent. I think the term "Pharmacological" means a lot of scientific aspects, and is far from the objectives of the present study. For this, it is more judicious to delete this word in the text.

Regarding the sample size, indeed we have used 07 mice per group. We are entirely of agreement that larger sample sizes are important to improve the robustness and reliability of the results. In our study, it was small size that due to budget restrictions. We could have used a large number in our experimental but sometimes we must adapt to field constraint. Additionally, we thought that the number seven is minimal samples, what it was doing in numerous studies. Further, we specify that all experiments have been doing in fully randomization procedures “male mice were randomly divided into four groups”.

In the present investigation, we suggest that the impaired sexual performance in male mice could be attributed to the action of bioactive compounds in the alkaloid extract. However, we believe that research will continue in the future to explain the mechanisms involved in the molecular action of the extract (please see the section "Conclusion"). This study remains a beginning of experimental work concerning the activity of P. harmala extract on reproductive function.

Reviewer 2 Report

The manuscript entitled "Effect of Peganumharmala total alkaloid extract on sexual behavior and sperm parameters in male mice" (Manuscript ID: vetsci-2474928) was designed to evaluate the effects of the total alkaloid extract of Peganum harmala Algerian seeds on sexual behavior and male reproductive function, and found that the extract affected the studied reproductive parameters of male mice in different manners. This is a very interesting study. However, the description of the results is not rigorous, there are subjective assumptions about the results, and the discussion lacks depth. At the same time, there are some formatting details. Therefore, for the benefit of the readers, the manuscript still needs major modifications before being accepted.

 Detailed comments are as follow:

1. The seeds of P. harmala were collected in April 2016 from Ngaoues, Batna province, whether the author took into account whether the storage time of seeds had an effect on the experimental results.

2. 6.25 mg/kg, 12.5 mg/kg and 25 mg/kg of P. harmala, how is the dose determined?

 3. at 48 hours and 6 hours, respectively. This way of writing is illogical

 4. Gonadosomatic Index (GSI) was estimated as a percentage of the total body weight relative to testicular weight.  This is wrong.

 5. The ratio of the number of motile sperm to the total number of sperm was used to indicate motility. Only the sperm that moves in a straight line is the sperm that has the ability to fertilize. The sperm motility calculation needs to use the sperm that moves in a straight line rather than the sperm that moves. The calculated sperm motility of this algorithm by the author is relatively high.

 6. Acronyms are not recommended because some words appear only once, for example, HRP. Some words use abbreviations without a definition, for example, CMC. There are also cases where acronyms are repeatedly defined, such as TAC. It is suggested that the author proofread and revise carefully.

 7. The results indicated that the extract of P. harmala at 25 mg/kg concentration seems to be more to influence sexual behavior in male mice., Regarding the IL, the male mice treated with different concentrations of total alkaloid extract of P. harmala had recorded a higher time than the control group. These two sentences are very loose in the description of the result. The author can not make subjective judgment, and it needs to be supported by strict data.

 8. The administration of total alkaloid P. harmala at any dose did not change the epididymis and seminal vesicle weight of the male mice. How does the author interpret the data of 0.082 ± 0.004*?

 9. As for the content of oxidative stress, the author only made a TAC, the research in this part is a little thin, and can not well illustrate the state of oxidative stress.

 10. Authors should discuss the results and how they can be interpreted from the perspective of previous studies and of the working hypotheses. This form of statement should not be included in the discussion.

 11. In the discussion, the author used a lot of space to describe the research results of others, but the literature cited and the information provided could not well support the results of this study.

 12. In their results, the author notes, the administration of total alkaloid P. harmala at any dose did not change the epididymis and seminal vesicle weight of the male mice. In the discussion, the author mentions, our results showed a rise in the weight of the epididymis and the GSI in male mice treated with P. harmala extracts. This is obviously contradictory, and authors should respect their own experimental results.

 13. In the discussion, the author mainly compared his own experimental results with those of others, and did not conduct in-depth analysis of his own experimental results, so the depth of the discussion was seriously insufficient.

 14. The conclusion is ambiguous.

Some formatting details need to be modified.

Author Response

Response to Reviewer #2 Comments

The manuscript entitled "Effect of Peganum harmala total alkaloid extract on sexual behavior and sperm parameters in male mice" (Manuscript ID: vetsci-2474928) was designed to evaluate the effects of the total alkaloid extract of Peganum harmala Algerian seeds on sexual behavior and male reproductive function, and found that the extract affected the studied reproductive parameters of male mice in different manners. This is a very interesting study. However, the description of the results is not rigorous, there are subjective assumptions about the results, and the discussion lacks depth. At the same time, there are some formatting details. Therefore, for the benefit of the readers, the manuscript still needs major modifications before being accepted.

Detailed comments are as follow:

Point 1: The seeds of P. harmala were collected in April 2016 from Ngaoues, Batna province, whether the author took into account whether the storage time of seeds had an effect on the experimental results.

Response 1:

We would like to inform the reviewer that the seeds of P. harmala were immediately extracted. Then the plant extract has been well stored in airtight bottles and in the absence of light until the experimental.

Point 2: 6.25 mg/kg, 12.5 mg/kg and 25 mg/kg of P. harmala, how is the dose determined?

Response 2:

The alkaloids in P. harmala are known to have toxic effects. According to the literature, the alkaloids of the P.harmala plant grown in Algeria are moderately toxic (lethal dose 50%: 350 mg/kg BW). previous studies have all focused on high concentrations. Our approach was based on the results of in-vitro effect of P. harmala extract on spermatozoa quality 'please see the reference below). This investigation demonstrated that sperm motility considerably increases at low concentrations of alkaloid extracts of P. harmala. Also, P. harmala alkaloids could protect sperm against oxidative damages and improve the membrane integrity of ram epididymal sperm. Therefore we have chosen these relatively low concentrations.

Benbott, A., Bahri, L., Boubendir, A., & Yahia, A. (2013). Study of the chemical components of Peganum harmala and evaluation of acute toxicity of alkaloids extracted in the Wistar albino mice. Journal of Material and Environmental Science4, 558-565.

Derbak, H., Moussaoui, M., Benberkane, A., Ayad, A. (2021). In-vitro effect of Peganum harmala total alkaloids on spermatozoa quality and oxidative stress of epididymal ram semen. Asian Pacific Journal of Reproduction, 10(5).

Point 3: at 48 hours and 6 hours, respectively. This way of writing is illogical.

Response 3:

It is correct. The female mice were made sexually receptive by administering estradiol benzoate 48 h, and progesterone 4 h before the Male sexual behavior test. Here below some references.

-Emmanuel, E. G., Sunday, U. A., & Sunda, T. P. (2020). Effect of methanol seed extract of Mucuna urens (l) medic on sexual behaviour and sperm parameters in male albino wistar rats. GSC Biological and Pharmaceutical Sciences, 11(1), 148-156.

-Al-Snafi, A. E. (2019). Chemical constituents and pharmacological effects of lepidium sativum. Int J Curr Pharm Res, 11(6), 1-10.

-Francine, M. M., Danielle, B. C., Desire, D. D. P., Mireille, K. P., Ngoungoure, M. C., Theophile, D., & Kamtchouing, P. (2017). Effects of Piper umbellatum linn.(piperaceae) leaves extract on aluminium chloride reproductive toxicity in male rats. WJPPS, 6(6), 338-362.

-Molina-Jiménez, T., Jiménez-Tlapa, M., Brianza-Padilla, M., Zepeda, R. C., Hernández-González, M., & Bonilla-Jaime, H. (2019). The neonatal treatment with clomipramine decreases sexual motivation and increases estrogen receptors expression in the septum of male rats: Effects of the apomorphine. Pharmacology Biochemistry and Behavior, 180, 83-91.

Point 4: Gonadosomatic Index (GSI) was estimated as a percentage of the total body weight relative to testicular weight.  This is wrong.

Response 4:

We would like to thank you for this comment; it was a mistake of formulation. It corrected in the text. “The Gonadosomatic Index was calculated by using formula that determined by testis weight divided by body weight”, this is ration.

Point 5: The ratio of the number of motile sperm to the total number of sperm was used to indicate motility. Only the sperm that moves in a straight line is the sperm that has the ability to fertilize. The sperm motility calculation needs to use the sperm that moves in a straight line rather than the sperm that moves. The calculated sperm motility of this algorithm by the author is relatively high.

Response 5:

Indeed, you are right, the linearity of sperm is important for its fertilizing ability. However, our goal in analyzing sperm mobility is not only the study of its fertilizing ability, it was to explore the quality of sperm taking into account the mobility in general and not only the linearity of sperm.

Point 6: Acronyms are not recommended because some words appear only once, for example, HRP. Some words use abbreviations without a definition, for example, CMC. There are also cases where acronyms are repeatedly defined, such as TAC. It is suggested that the author proofread and revise carefully.

Response 6:

Point 7: “The results indicated that the extract of P. harmala at 25 mg/kg concentration seems to be more to influence sexual behavior in male mice.”, “Regarding the IL, the male mice treated with different concentrations of total alkaloid extract of P. harmala had recorded a higher time than the control group”. These two sentences are very loose in the description of the result. The author cannot make subjective judgment, and it needs to be supported by strict data.

Response 7:

It is noted. We modified in the revised version.

Point 8: The administration of total alkaloid P. harmala at any dose did not change the epididymis and seminal vesicle weight of the male mice. How does the author interpret the data of 0.082 ± 0.004*?

Response 8:

Absolutely, the administration of total alkaloid P. harmala at any dose did not modify the seminal vesicles weight of the treated male mice. The values are almost similar and not significant.

On the other hand, the epididymis weight of the treated male mice with total alkaloid P. harmala concentrations 25 mg/kg and 6.25 mg/kg did not change, but the treated group with 12.5 mg/kg concentration presented à significant (P < 0.05) light increase of epididymis weight (0.082 ± 0.004) compared to control group (0.063 ± 0.001).

Point 9: As for the content of oxidative stress, the author only made a TAC, the research in this part is a little thin, and can not well illustrate the state of oxidative stress.

Response 9:

Total antioxidant capacity is an overall measure of the body's ability to defend itself against ROS. It reflects the synergy that can exist between the different components of the antioxidant system. by carrying out this test we obtained some preliminary results. I would like to inform you that the study is not yet complete and that we are going to carry out other tests to investigate oxidative stress, using other tests such as TBARs..., etc.

We completed in the revised version some information about oxidative status in sperm.

-Valko M, Leibfritz D, Moncol J, Cronin MT, Mazur M, Telser J. Free radicals and antioxidants in normal physiological functions and human disease. Int J Biochem Cell Biol. 2007;39(1):44–84.

-Hosseinzadeh Colagar A, Karimi F, Jorsaraei SG. Correlation of sperm parameters with semen lipid peroxidation and total antioxidants levels in astheno- and oligoasheno- teratospermic men. Iran Red Crescent Med J. 2013;15(9):780–5.

-Fazeli, F., & Salimi, S. (2016). Correlation of seminal plasma total antioxidant capacity and malondialdehyde levels with sperm parameters in men with idiopathic infertility. Avicenna J Med Biochem.; 4(1):e29736.

-Sarbishegi, M., Gorgich, E. A. C., & Khajavi, O. (2017). Olive leaves extract improved sperm quality and antioxidant status in the testis of rat exposed to rotenone. Nephro-Urology Monthly, 9(3). :e47127.

Point 10: Authors should discuss the results and how they can be interpreted from the perspective of previous studies and of the working hypotheses. This form of statement should not be included in the discussion. Point 11: In the discussion, the author used a lot of space to describe the research results of others, but the literature cited and the information provided could not well support the results of this study.

Response 10 and 11:

We corrected it in the revised version.

Point 12: In their results, the author notes, the administration of total alkaloid P. harmala at any dose did not change the epididymis and seminal vesicle weight of the male mice. In the discussion, the author mentions, our results showed a rise in the weight of the epididymis and the GSI in male mice treated with P. harmala extracts. This is obviously contradictory, and authors should respect their own experimental results.

Response 12:

We revised this point in the text. Please, see the response of the point 9

Point 13: In the discussion, the author mainly compared his own experimental results with those of others, and did not conduct in-depth analysis of his own experimental results, so the depth of the discussion was seriously insufficient.

Response 13:

We corrected it in the revised version.

Point 14: The conclusion is ambiguous.

Response 14: 

Point 15: Some formatting details need to be modified.

Response 15:

Reviewer 3 Report

Dear authors,

I have read the submitted paper with great interest. Infertility issues are a significant problem in both humans and animals today, and therefore, they are always a subject of research. However, when it comes to fertilization problems, both male and female partners are involved. The parameters concerning female reproductive parameters (plugging rate, plugging quality, fertilization rate, etc.) were completely missing. The females were synchronized with estrogen and progesterone to analyze the reproductive parameters of the males, but what happened to the females? It appears that the males were not vasectomized, so offspring must have been produced?

You performed many assays, but I find a lack of investigation into the presumed mechanism of action.

It is an interesting topic, but it requires thorough revision.

The English text needs significant revision, for instance the first sentence.

Author Response

Response to Reviewer #3 Comments

Point 1: I have read the submitted paper with great interest. Infertility issues are a significant problem in both humans and animals today, and therefore, they are always a subject of research. However, when it comes to fertilization problems, both male and female partners are involved. The parameters concerning female reproductive parameters (plugging rate, plugging quality, fertilization rate, etc.) were completely missing. The females were synchronized with estrogen and progesterone to analyze the reproductive parameters of the males, but what happened to the females? It appears that the males were not vasectomized, so offspring must have been produced?

Response 1:

We are entirely of agreement. The infertility is a serious problem in animal reproduction and also in human couples. It’s a current subject and will remain it. Our investigation is focused mainly on the affect the P. harmaala extract on reproductive function only in male mice. We think also it is a good idea to undertake in perspective to study the fecundation aspect in female, especially the fertilization rate, number of new born...).

We have used a standard protocol of synchronization base on estradiol and progesterone to evaluate the reproductive parameters in male. Then the females were excluded of the experiments, without to be following the pregnancy without following the pregnancy status of the female mice. In addition, the males were not vasectomized.

Point 2: You performed many assays, but I find a lack of investigation into the presumed mechanism of action.

Response 2:

This research can be considered a preliminary study which will pave the way for further in-depth research into the mechanism of action of Peganum harmala extract on sexual behaviour and the male reproductive system. We think sincerely it is a good idea to undertake in future in order to understand the mechanism of action of bioactive molecular on the reproductive function. This is presented in the section "conclusion" as a perspective of research.

Point 3: It is an interesting topic, but it requires thorough revision. The English text needs significant revision, for instance the first sentence. 

Response 3: 

We corrected the English language in the revised version.

Reviewer 4 Report

Manuscript vetsi-2474928

Introduction:

It's not clear why the authors believe that P. harmala could have some influence on man's reproductive health well. Why do the authors think this plant can influence male reproductive function? In lines 75-80, the authors mention that this plant has been used as an anti-inflammatory, antifungal, insecticidal, antibacterial, and antidiabetic. Moreover, in lines 82-84, the authors mention that there are few studies about the use of this plant in man reproductive disorders. So, my question is about how the authors believe that this plant can be used to improve reproductive health in men.

Material and Methods:

I don't have any suggestions. I consider that to be well-written.

Results:

I don't have any suggestions. I believe that to be well-written.

Discussion:

Lines 276 -277: Authors wrote, "Authors should discuss the results and how they can be interpreted from the per- 276 spective of previous studies and of the working hypotheses" This sentence is part of the MDPI paper template. Please delete it.

Line 304: Add a space between "thatserotonin"

The authors should discuss the results in Figure 2 corresponding to the spermicide effect that had treatments of 12.5 mg/kg and 6.25 mg/kg. Why did a lower P. harmala concentration decreases the sperm count? How can the authors explain that a higher dose of P. harmala increases sperm count? These results are contradictory.

Author Response

Response to Reviewer #4 Comments

Point 1: Introduction: It's not clear why the authors believe that P. harmala could have some influence on man's reproductive health well. Why do the authors think this plant can influence male reproductive function? In lines 75-80, the authors mention that this plant has been used as an anti-inflammatory, antifungal, insecticidal, antibacterial, and antidiabetic. Moreover, in lines 82-84, the authors mention that there are few studies about the use of this plant in man reproductive disorders. So, my question is about how the authors believe that this plant can be used to improve reproductive health in men.

Response 1:

Point 2: Material and Methods: I don't have any suggestions. I consider that to be well-written.

Point 3: Results: I don't have any suggestions. I believe that to be well-written.

Response 2:

Thank you for this comment.

Point 4: Lines 276 -277: Authors wrote, "Authors should discuss the results and how they can be interpreted from the per. Point 5: Line 276 spective of previous studies and of the working hypotheses" This sentence is part of the MDPI paper template. Please delete it.

Response 4 and 5:

Thank you very much for this comment. It was a redaction error when we prepared the submission manuscript the Microsoft we have used the Word template which contain this sentence in section "Discussion". We have removed this sentence in the revised version. We apologize for this.

Point 6: Line 304: Add a space between "thatserotonin"

Response 6:

Thank you for this remark. It’s corrected in the revised version.

Point 7: The authors should discuss the results in Figure 2 corresponding to the spermicide effect that had treatments of 12.5 mg/kg and 6.25 mg/kg. Why did a lower P. harmala concentration decreases the sperm count? How can the authors explain that a higher dose of P. harmala increases sperm count? These results are contradictory.

Response 7:

Indeed, the sperm count results seem inexplicable because of this difference between high and low concentration treatments of P. harmala. We could say is at concentration 25 mg/kg tends to protect sperm cells, therefore their number are higher compared to treatment group at low concentrations (12.5 mg/kg and 6.25 mg/kg). We propose to undertake another in vitro study to answer this hypothesis of spermicidal or protective effect of spermatozoids with different concentrations.

Round 2

Reviewer 1 Report

The authors have improved the manuscript sufficiently. Now it is suitable for publication. 

Author Response

Response to Reviewer #1 Comments

The authors have improved the manuscript sufficiently. Now it is suitable for publication.

Dear reviewer, our sincere thanks for taking the time to review this manuscript, and we are delighted that you are satisfied by our answers. We highly appreciate your overall positive feed-back regarding the quality of the manuscript, giving us the chance to revise it!

Thank you again!

Reviewer 2 Report

Firstly, the questions raised by the reviewer are a problem that all readers face. Therefore, when the author answers the reviewer's questions, they are not meant to explain to the reviewer, but to make corresponding modifications to the manuscript. This means that answering the reviewer's questions needs to be reflected in the manuscript. Unfortunately, the author did not do so and instead provided lengthy explanations to the reviewer. Secondly, the author did not fully answer the questions raised by the reviewers, such as Point 6, Point 14, and Point 15 Finally, this study is not complete, as the author puts it, "I would like to inform you that the study is not yet complete and that we are going to carry out other tests to investigate oxidative stress, using other tests such as TBARs..., etc." It is recommended that the author improve the experiment before submitting.

Author Response

Response to Reviewer #2 Comments

Firstly, the questions raised by the reviewer are a problem that all readers face. Therefore, when the author answers the reviewer's questions, they are not meant to explain to the reviewer, but to make corresponding modifications to the manuscript. This means that answering the reviewer's questions needs to be reflected in the manuscript. Unfortunately, the author did not do so and instead provided lengthy explanations to the reviewer.

Secondly, the author did not fully answer the questions raised by the reviewers, such as Point 6, Point 14, and Point 15 Finally, this study is not complete, as the author puts it, "I would like to inform you that the study is not yet complete and that we are going to carry out other tests to investigate oxidative stress, using other tests such as TBARs..., etc." It is recommended that the author improve the experiment before submitting.

Thank you for reminding us of these points. I reassure you that we had already answered these questions in the previous revision but by mistake they were not inserted in the file sent. We apologize for this inconvenience.

Point 2:6.25 mg/kg, 12.5 mg/kg and 25 mg/kg of P. harmala, how is the dose determined? Response 2: The alkaloids in P. harmala are known to have toxic effects. According to the literature, the alkaloids of the P.harmala plant grown in Algeria are moderately toxic (lethal dose 50%: 350 mg/kg BW). Previous studies have all focused on high concentrations. Our approach was based on the results of in-vitro effect of P. harmala extract on spermatozoa quality 'please see the reference below). This investigation demonstrated that sperm motility considerably increases at low concentrations of alkaloid extracts of P. harmala. Also, P. harmala alkaloids could protect sperm against oxidative damages and improve the membrane integrity of ram epididymal sperm. Therefore we have chosen these relatively low concentrations.

Benbott, A., Bahri, L., Boubendir, A., & Yahia, A. (2013). Study of the chemical components of Peganum harmala and evaluation of acute toxicity of alkaloids extracted in the Wistar albino mice. Journal of Material and Environmental Science4, 558-565.

Derbak, H., Moussaoui, M., Benberkane, A., Ayad, A. (2021). In-vitro effect of Peganum harmala total alkaloids on spermatozoa quality and oxidative stress of epididymal ram semen. Asian Pacific Journal of Reproduction, 10(5).

Point 3: at 48 hours and 6 hours, respectively. This way of writing is illogical. Response 3: The female mice were made sexually receptive by administering estradiol benzoate 48 h, and progesterone 4 h before the Male sexual behavior test. Here below some references. We corrected it in the revised version.  Please, see the line 145 of the revised version.

-Emmanuel, E. G., Sunday, U. A., & Sunda, T. P. (2020). Effect of methanol seed extract of Mucuna urens (l) medic on sexual behaviour and sperm parameters in male albino wistar rats. GSC Biological and Pharmaceutical Sciences, 11(1), 148-156.

-Al-Snafi, A. E. (2019). Chemical constituents and pharmacological effects of lepidium sativum. Int J Curr Pharm Res, 11(6), 1-10.

-Francine, M. M., Danielle, B. C., Desire, D. D. P., Mireille, K. P., Ngoungoure, M. C., Theophile, D., & Kamtchouing, P. (2017). Effects of Piper umbellatum linn.(piperaceae) leaves extract on aluminium chloride reproductive toxicity in male rats. WJPPS, 6(6), 338-362.

-Molina-Jiménez, T., Jiménez-Tlapa, M., Brianza-Padilla, M., Zepeda, R. C., Hernández-González, M., & Bonilla-Jaime, H. (2019). The neonatal treatment with clomipramine decreases sexual motivation and increases estrogen receptors expression in the septum of male rats: Effects of the apomorphine. Pharmacology Biochemistry and Behavior, 180, 83-91.

Point 6: Acronyms are not recommended because some words appear only once, for example, HRP. Some words use abbreviations without a definition, for example, CMC. There are also cases where acronyms are repeatedly defined, such as TAC. It is suggested that the author proofread and revise carefully. Response 6: Thank you for this remark. We corrected and checked it in the revised version. Please, see the line 139, 195, 284 of the revised version

Point 8:The administration of total alkaloid P. harmala at any dose did not change the epididymis and seminal vesicle weight of the male mice. How does the author interpret the data of 0.082 ± 0.004*? Response 8: Absolutely, the administration of total alkaloid P. harmala at any dose did not modify the seminal vesicles weight of the treated male mice. The values are almost similar and not significant.

On the other hand, the epididymis weight of the treated male mice with total alkaloid P. harmala concentrations 25 mg/kg and 6.25 mg/kg did not change, but the treated group with 12.5 mg/kg concentration presented à significant (P < 0.05) light increase of epididymis weight (0.082 ± 0.004) compared to control group (0.063 ± 0.001).

Point 9: As for the content of oxidative stress, the author only made a TAC, the research in this part is a little thin, and can not well illustrate the state of oxidative stress. Response 9: This aspect can be considered as study limitation, having a reference effect in stimulating further opportunities for studies to be carried out to address this, as I highlighted in my revision note draft. The research team doesn’t have presently the financial support to do supplementary investigations

Total antioxidant capacity is an overall measure of the body's ability to defend itself against ROS. It reflects the synergy that can exist between the different components of the antioxidant system by carrying out this test we obtained some preliminary results. I would like to inform you that the study is not yet complete and that we are going to carry out other tests to investigate oxidative stress, using other tests such as TBARs..., etc. We completed in the revised version some information about oxidative status in sperm.

-Valko M, Leibfritz D, Moncol J, Cronin MT, Mazur M, Telser J. Free radicals and antioxidants in normal physiological functions and human disease. Int J Biochem Cell Biol. 2007;39(1):44–84.

-Hosseinzadeh Colagar A, Karimi F, Jorsaraei SG. Correlation of sperm parameters with semen lipid peroxidation and total antioxidants levels in astheno- and oligoasheno- teratospermic men. Iran Red Crescent Med J. 2013;15(9):780–5.

-Fazeli, F., & Salimi, S. (2016). Correlation of seminal plasma total antioxidant capacity and malondialdehyde levels with sperm parameters in men with idiopathic infertility. Avicenna J Med Biochem.; 4(1):e29736.

-Sarbishegi, M., Gorgich, E. A. C., & Khajavi, O. (2017). Olive leaves extract improved sperm quality and antioxidant status in the testis of rat exposed to rotenone. Nephro-Urology Monthly, 9(3). :e47127.

Point 14: The conclusion is ambiguous. Response 14: In the author’s opinion the present form of the conclusion section properly serves to help the reader to understand why the results of the present study should matter to them.

Point 15: Some formatting details need to be modified. Response 15: We corrected it in the revised version. Please, see the line from 417 to 427 of the revised version

Reviewer 3 Report

Dear authors,

I have reread the presented paper, and it has significantly improved. However, in my opinion, there are still some issues, that need a revision before publication.

  1. In line 27, you mention that no change in sperm motility was observed in the treated groups, but in line 31, you state that it can have an effect on the motility and count of sperm. Please clarify this discrepancy.
  2. In line 51, for the group treated with 25 mg/kg, the sperm count was stable. Could you provide an explanation for why there is a decrease in groups treated with a lower amount?
  3. In line 70, you mention that around 50% of infertility cases are of male origin, often due to their inability to fertilize. Therefore, I find it challenging to understand a treatment focused on males without addressing whether any offspring were born at all.
  4. In line 102, are there basic parameters about the strain used for the investigation of sexual behavior? Since most mouse strains exhibit good libido that decreases with age, providing basic details about the strain is essential.
  5. In line 33, could you specify the age of the mice weighing 30g?
  6. In line 331, you state that dopamine is an excitable neurotransmitter responsible for the motivated aspect of sexual behavior. However, in line 338, you mention that P. harmala is suspected to be a dopamine receptor antagonist, isn’t this contradictory for the effect of P. hamala?

Thank you for addressing these concerns before finalizing the paper.Formularbeginn

Author Response

Response to Reviewer #3 Comments

Dear authors,

I have reread the presented paper, and it has significantly improved. However, in my opinion, there are still some issues, that need a revision before publication.

Dear reviewer, our sincere thanks for taking the time to review the revised version of the manuscript, and your close attention to detail. We highly appreciate your overall positive feed-back regarding the quality of the revision, giving us the chance to revise it! All the remained concerns were carefully approached. Please see below for our responses to your comments:

  1. In line 27, you mention that no change in sperm motility was observed in the treated groups, but in line 31, you state that it can have an effect on the motility and count of sperm. Please clarify this discrepancy. Answer: It is right. The total alkaloid P. harmala extract affects the gonadosomatic index and sperm count. We corrected it in the revised version. Please, see the line 31 of the revised version.

  1. In line 51, for the group treated with 25 mg/kg, the sperm count was stable. Could you provide an explanation for why there is a decrease in groups treated with a lower amount? Answer: The decrease in the concentration of semen from the groups treated with the lowest concentrations compared with the concentration of semen from the group treated with the highest concentration (25 mg/kg), which was stable, is probably due to the dilution medium because the lowest concentrations have more distilled water than the highest concentration. However, this result requires further exploration (in future).

  1. In line 70, you mention that around 50% of infertility cases are of male origin, often due to their inability to fertilize. Therefore, I find it challenging to understand a treatment focused on males without addressing whether any offspring were born at all. Answer: Exactly, to incriminate the male as a cause of infertility, it would be necessary to see if there is an offspring born from him. This sentence "about 50% of cases of infertility are of male origin ..." is an observation reported by Saleem et al. (2020), we want to emphasize that the man is also part of the problem of infertility.

We hypothesize that our extract could have a direct effect on the birth rate, since this parameter is directly correlated with sperm quality and sexual vigor, which increased significantly in our study. However, this hypothesis would need to be translated into a full study.

  1. In line 102, are there basic parameters about the strain used for the investigation of sexual behavior? Since most mouse strains exhibit good libido that decreases with age, providing basic details about the strain is essential. Answer: We used Mus musculus Swiss strain of mice as they are available and versatile mice used in all areas of biomedical research. From a physiological point of view, the strain has a good libido which makes it a good model for the study of sexual behaviour. Moreover, this work was carried out on the same type of strain which has no effect on the study results. We added strain of mice in the revised version (Please, see the line 105 and 135 of the revised version.

  1. In line 33, could you specify the age of the mice weighing 30g? Answer: The mice aged 8 to 10 weeks. Please, see the line 131 of the revised version.

  1. In line 331, you state that dopamine is an excitable neurotransmitter responsible for the motivated aspect of sexual behavior. However, in line 338, you mention that P. harmala is suspected to be a dopamine receptor antagonist, isn’t this contradictory for the effect of P. hamala? Answer: We agree with you. Thank you very much for this comment. We changed it in the revised version. Please, see the line from 330 to 333 of the revised version. “Indeed, the effects of β-carbolines on dopaminergic transmission have been the subject of neurochemical and behavioural studies. These have shown that certain β-carboline alkaloids can facilitate dopaminergic transmission and interact with dopamine D1 and D2 receptors in the brain [41].” [41] Patel, K., Gadewar, M., Tripathi, R., Prasad, S. K., & Patel, D. K.. A review on medicinal importance, pharmacological activity and bioanalytical aspects of beta-carboline alkaloid “Harmine”. Asian Pacific journal of tropical biomedicine 2012; 2(8), 660-664.

Thank you for addressing these concerns before finalizing the paper.

Thank you again!

Reviewer 4 Report

I consider this document a good material to be published.

Author Response

Response to Reviewer #4 Comments

I consider this document a good material to be published.

Answer: Dear reviewer, our sincere thanks for taking the time to review this manuscript, and we are delighted that you are satisfied by our answers. We highly appreciate your overall positive feed-back regarding the quality of the manuscript, giving us the chance to revise it!

Thank you very much again!

Round 3

Reviewer 2 Report

The author has carefully revised the manuscript according to the suggestions of the reviewers, and has answered every question, so that the manuscript can be considered for publication.